# ANNEALED IMPORTANCE SAMPLING MEETS SCORE MATCHING

**Arnaud Doucet, Will Grathwohl, Alex G.D.G. Matthews & Heiko Strathmann** [*]
DeepMind
{arnauddoucet,wgrathwohl,alexmatthews,strathmann}@google.com

## ABSTRACT

Annealed Importance Sampling (AIS) is one of the most effective methods for marginal likelihood estimation. It relies on a sequence of distributions interpolating between a tractable initial distribution and the posterior of interest which we simulate from approximately using a non-homogeneous Markov chain. To obtain an importance sampling (IS) estimate of the marginal likelihood, AIS introduces an extended target distribution to reweight the Markov chain proposal. While much effort has been devoted to improving the proposal distribution used by AIS by changing the intermediate distributions and corresponding Markov kernels, an underappreciated issue is that AIS uses an convenient but suboptimal extended target distribution which can hinder its performance. We leverage here recent progress in score-based generative modeling to learn the optimal extended target distribution for a given AIS proposal using score matching ideas. We demonstrate this novel differentiable AIS procedure on a number of synthetic benchmark distributions and a normalizing flow target.

## 1 INTRODUCTION

Evaluating the marginal likelihood, also known as evidence, is of key interest in Bayesian statistics as it allows not only model comparison but is also often used to select hyperparameters. A large variety of Monte Carlo methods have been proposed to address this problem, including path sampling (Gelman & Meng, 1998), AIS (Neal, 2001) and related SMC methods (Del Moral et al., 2006). An appealing feature of AIS is that it provides an unbiased estimate of the marginal likelihood and can thus be used to define an evidence lower bound (ELBO); see e.g. (Wu et al., 2020; Thin et al., 2021).

AIS builds a proposal distribution using a Markov chain $(x_k)_{k=0}^K$ initialized at an easy-to-sample distribution followed by a sequence of Markov chain Monte Carlo (MCMC) transitions targeting typically annealed versions of the posterior. By proceeding this way, we obtain a proposal $x_K$ whose distribution is expected to be a reasonable approximation to the posterior. However, this distribution is intractable as it requires integrating the joint proposal distribution over previous states $(x_k)_{k=0}^{K-1}$. AIS bypasses this issue by instead using IS on the whole path $(x_k)_{k=0}^K$ through the introduction of an artificial extended target distribution whose marginal at time $K$ coincides with the posterior.

There has been much work devoted to improving AIS by modifying the intermediate distributions (Grosse et al., 2013; Brekelmans et al., 2020) and corresponding transition kernels of the proposal (Dai et al., 2020; Wu et al., 2020; Geffner & Domke, 2021; Thin et al., 2021; Zhang et al., 2021). We here address a distinct issue: it was shown in (Del Moral et al., 2006) that the extended target distribution used by AIS is suboptimal and that the optimal extended target minimizing the variance of the evidence estimate is defined through the time-reversal of the proposal. This result has never been exploited algorithmically as the time-reversal is intractable. Inspired by recent developments on score-based generative modeling (Ho et al., 2020; Song et al., 2021b) which approximates the time-reversal of a noising diffusion process, we parameterize an approximation to the time-reversal which we learn by maximizing an ELBO. As in score-based generative modeling, this ELBO is shown to coincide with a denoising score matching loss (Vincent, 2011; Ho et al., 2020; Song et al., 2021b). This provides a novel, optimized and differentiable, AIS estimator. We demonstrate the

---

[*]alphabetical order, equal contribution.

benefits of this approach on various synthetic benchmark distributions and a normalizing flow target. All proofs can be found in the Appendix.

## 2 ANNEALED IMPORTANCE SAMPLING

**Setup and algorithm.** Let $\pi(x) = \gamma(x)/Z$ be a density on $\mathbb{R}^d$ where $\gamma(x)$ can be evaluated and we want to approximate its intractable normalizing constant $Z = \int \gamma(x)\mathrm{d}x$. In a Bayesian framework, $\gamma(x) = p(x)p(\mathcal{D}|x)$ is the joint density of parameter $x$ and data $\mathcal{D}$, $\pi(x) = p(x|\mathcal{D})$ the corresponding posterior and $Z = p(\mathcal{D})$ the evidence.

To estimate $Z$, AIS introduces the intermediate distributions $(\pi_k)_{k=1}^K$ bridging smoothly from a tractable distribution $\pi_0$ to the target distribution $\pi_K = \pi$ of interest. One typically uses $\pi_k(x) \propto \gamma_k(x)$ with $\gamma_k(x) = \pi_0(x)^{1-\beta_k}\gamma_K(x)^{\beta_k}$ for $0 = \beta_0 < \beta_1 < \cdots < \beta_K = 1$ but other choices are possible. The IS proposal used by AIS is then obtained by running a Markov chain $(x_k)_{k=0}^K$ such that $x_0 \sim \pi_0(\cdot)$, and then $x_k \sim F_k(\cdot|x_{k-1})$ for $k \geq 1$ where $F_k$ is a MCMC kernel invariant w.r.t. $\pi_k$. The proposal is thus given by

$$Q(x_{0:K}) = \pi_0(x_0) \prod_{k=1}^K F_k(x_k|x_{k-1}). \tag{1}$$

Denote by $q_k$ the intractable marginal distribution of $x_k$ under $Q$ satisfying $q_k(x_k) = \int q_{k-1}(x_{k-1})F_k(x_k|x_{k-1})\mathrm{d}x_{k-1}$ for $k \geq 1$ and $q_0 = \pi_0$. As $q_K$ cannot be evaluated, the marginal IS estimate $w_{\mathrm{mar}}(x_K) = \gamma_K(x_K)/q_K(x_K)$ of $Z$ is intractable. AIS bypasses this issue by introducing an extended target

$$P(x_{0:K}) = \Gamma(x_{0:k})/Z, \qquad \Gamma(x_{0:K}) = \gamma_K(x_K) \prod_{k=0}^{K-1} B_k(x_k|x_{k+1}), \tag{2}$$

where $(B_k)_{k=0}^{K-1}$ are backward Markov transition kernels. For any selection of backward kernels such that the ratio $\Gamma/Q$ is well-defined, $w(x_{0:K}) = \Gamma(x_{0:K})/Q(x_{0:K})$ is an unbiased estimate of $Z$ for $x_{0:K} \sim Q$. AIS relies on the specific choice $B_k^{\mathrm{ais}}(x'|x) = \pi_{k+1}(x')F_{k+1}(x|x')/\pi_{k+1}(x)$ which yields the AIS log-evidence estimate $\log w_{\mathrm{ais}}(x_{0:K}) = \sum_{k=1}^K \log(\gamma_k(x_{k-1})/\gamma_{k-1}(x_{k-1}))$.

**Limitations of AIS.** While designing $P$ in (2) by using $(B_k^{\mathrm{ais}})_{k=0}^{K-1}$ is convenient, it is clearly suboptimal. For example, consider the ideal scenario where $F_k(x'|x) = \pi_k(x')$ then $\mathrm{var}_Q[w_{\mathrm{ais}}(x_{0:K})] > 0$ while $\mathrm{var}_{q_K}[w_{\mathrm{mar}}(x_K)] = 0$. Another illustration of the suboptimality of AIS is to consider a scenario where the proposal is an homogeneous MCMC chain, i.e. $x_0 \sim \pi_0$ and $x_k \sim F(\cdot|x_{k-1})$ for $F$ a $\pi$-invariant MCMC kernel; i.e. use $F_k = F$ and $\pi_k = \pi$ for $k = 1, ..., K$. If $F$ is reasonably well-mixing, then $q_K \approx \pi$ for $K$ large enough and the evidence estimate $w_{\mathrm{mar}}(x_K) = \gamma_K(x_K)/q_K(x_K)$ should have small variance. However, we have $w_{\mathrm{ais}}(x_{0:K}) = \gamma(x_0)/\pi_0(x_0)$ for the exact same proposal; i.e. the AIS estimate does not depend on the MCMC samples $x_{1:K}$ and boils down to the IS estimate of $Z$ using the proposal $\pi_0$. These examples illustrate that it would be preferable to use $w_{\mathrm{mar}}(x_K)$ rather than $w_{\mathrm{ais}}(x_{0:K})$. We propose in the next section an unbiased estimate of the evidence approximating $w_{\mathrm{mar}}(x_K)$.

## 3 OPTIMIZED ANNEALED IMPORTANCE SAMPLING

We show here that the optimal extended target distribution $P$ is defined through the time-reversal of the proposal $Q$. By exploiting a connection to score-based generative models, we then approximate this reversal using score matching when the proposal is an unadjusted Langevin algorithm (ULA).

**Optimal Extended Target Distribution via Time Reversal.** We summarize here Proposition 1 of Del Moral et al. (2006); see also (Sohl-Dickstein et al., 2015).

**Proposition 1.** *For a proposal $Q$ of the form (1), the extended target $P$ of the form (2) minimizing both $D_{\mathrm{KL}}(Q||P)$ and the variance of the evidence estimate $w(x_{0:K}) = \Gamma(x_{0:K})/Q(x_{0:K})$ for $x_{0:K} \sim Q$ is given by $P_{\mathrm{opt}}(x_{0:K}) = \Gamma_{\mathrm{opt}}(x_{0:K})/Z$ where*

$$\Gamma_{\mathrm{opt}}(x_{0:K}) = \gamma_K(x_K) \prod_{k=0}^{K-1} B_k^{\mathrm{opt}}(x_k|x_{k+1}), \qquad B_k^{\mathrm{opt}}(x_k|x_{k+1}) = \frac{q_k(x_k)F_{k+1}(x_{k+1}|x_k)}{q_{k+1}(x_{k+1})}. \tag{3}$$

*In particular, one has $w_{\mathrm{mar}}(x_K) = \Gamma_{\mathrm{opt}}(x_{0:K})/Q(x_{0:K})$ and $D_{\mathrm{KL}}(Q||P_{\mathrm{opt}}) = D_{\mathrm{KL}}(q_K||\pi_K)$.*

We emphasize that Proposition 1 applies to any forward kernels $(F_k)_{k=1}^K$ including MCMC kernels, ULA kernels or even deterministic maps. It shows that $P^{\text{opt}}$ is the distribution of a backward process initialized at $\pi_K$ which then follows the time-reversed dynamics of the forward process $Q$.

**Time reversal, Score matching and ELBO.** We concentrate henceforth on the case where $(F_k)_{k=1}^K$ are ULA kernels as used in (Heng et al., 2020; Wu et al., 2020; Thin et al., 2021); that is we consider $F_k(x'|x) = \mathcal{N}(x'; x + \delta \nabla \log \pi_k(x), 2\delta I)$ where $\delta > 0$ is a stepsize. Let $\delta := T/K$ then, as $K \to \infty$, the proposal $Q$ converges to the path measure $\mathcal{Q}$ of the following inhomogeneous Langevin diffusion $(X_t)_{t \in [0,T]}$

$$\mathrm{d}X_t = \nabla \log \pi_t(X_t)\mathrm{d}t + \sqrt{2}\mathrm{d}B_t, \qquad X_0 \sim \pi_0, \tag{4}$$

where $(B_t)_{t \in [0,T]}$ is the standard multivariate Brownian motion and we are slightly abusing notation as $\pi_t$ for $t = t_k = k\delta$ corresponds to $\pi_k$ in discrete-time for $\delta = T/K$. The time-reversed process $(Y_t) = (X_{T-t})_{t \in [0,T]}$ is also a diffusion (Haussmann & Pardoux, 1986)

$$\mathrm{d}Y_t = \big\{ -\nabla \log \pi_{T-t}(Y_t) + 2\nabla \log q_{T-t}(Y_t)\big\}\mathrm{d}t + \sqrt{2}\mathrm{d}B_t, \qquad Y_0 \sim q_T. \tag{5}$$

The continuous-time version of $P_{\text{opt}}$ is the path measure $\mathcal{P}_{\text{opt}}$ defined by the diffusion (5) but initialized at $Y_0 \sim \pi_T$ rather than $q_T$ as noted in Bernton et al. (2019). This shows that approximating $(B_k^{\text{opt}})_{k=0}^{K-1}$ requires approximating the so-called scores $(\nabla \log q_t(x))_{t \in [0,T]}$, which are the continuous-time versions of the ratios $q_{k+1}(x_{k+1})/q_k(x_k)$ appearing in $B_k^{\text{opt}}(x_k|x_{k+1})$.

In score-based generative models (Song et al., 2021b), a powerful class of models that has become recently very popular, one gradually adds noise to data using an Ornstein–Ulhenbeck process and the generative model is obtained by approximating the time-reversal of this diffusion initialized by Gaussian noise. Practically, the time-reversal approximation is obtained by estimating the scores of the noising diffusion using denoising score matching (Vincent, 2011). We here adapt this idea to our setup. We define a path measure $\mathcal{P}_\theta$ by plugging a neural network $s_\theta(T - t, Y_t)$ in place of $\nabla \log q_{T-t}(Y_t)$ in (5). We learn $\theta$ by minimizing $D_{\text{KL}}(\mathcal{Q}||\mathcal{P}_\theta)$ over $\theta$, i.e. maximize a continuous-time ELBO, which we will show below coincides with a score matching loss as in the generative modeling context (Song et al., 2021a). Note that it is neither easily feasible to minimize $D_{\text{KL}}(\mathcal{P}^{\text{opt}}||\mathcal{P}_\theta)$ as one cannot sample from $\pi_T$ nor it is desirable as the evidence estimate is computed using samples from the proposal. Practically the diffusions corresponding to $\mathcal{Q}$ and $\mathcal{P}_\theta$ have to be discretized so a more direct route is to simply take inspiration of (5) and consider $B_k^\theta(x'|x) = \mathcal{N}(x'; x - \delta \nabla \log \pi_{k+1}(x) + 2\delta s^\theta(k+1, x), 2\delta I)$ to obtain a parameterized extended target $P_\theta$ and corresponding unnormalized target $\Gamma_\theta$ and learn $\theta$ by minimizing $D_{\text{KL}}(Q||P_\theta)$. These two approaches coincide for $\delta \ll 1$ as shown below. Once $\theta$ is learned, we can then estimate unbiasedly the evidence through $w_\theta(x_{0:K}) = \Gamma_\theta(x_{0:K})/Q(x_{0:K})$ for $x_{0:K} \sim Q(\cdot)$.

**Proposition 2.** *Under regularity conditions, we have*

$$D_{\text{KL}}(\mathcal{Q}||\mathcal{P}_\theta) = \mathbb{E}_{\mathcal{Q}}\Big[\int_0^T ||s_\theta(t, X_t) - \nabla \log q_t(X_t)||^2 \mathrm{d}t\Big] + C_1$$

$$= \sum_{k=1}^K \int_{t_{k-1}}^{t_k} \mathbb{E}_{\mathcal{Q}}\left[||s_\theta(t, X_t) - \nabla \log q_{t|t_{k-1}}(X_t|X_{t_{k-1}})||^2\right]\mathrm{d}t + C_2, \tag{6}$$

*where $t_k = k\delta$, $K = T/\delta$, $q_{t|s}(x'|x)$ is the density of $X_t = x'$ given $X_s = x$ under $\mathcal{Q}$ and $C_1, C_2$ constants independent of $\theta$. Let $\mathcal{L}(\theta) = \delta \sum_{k=1}^K \mathbb{E}_Q\left[||s_\theta(k, x_k) - \nabla \log F_k(x_k|x_{k-1})||^2\right]$ denote a discrete-time approximation of this loss. We have $\nabla D_{\text{KL}}(Q||P_\theta) = \nabla \mathcal{L}(\theta) + \epsilon(\theta)$ for some function $\epsilon$ satisfying $\lim_{K \to \infty} \epsilon(\theta) = 0$.*

## 4 EXPERIMENTS

We run a number of experiments where we estimate normalizing constants to validate our approach, Monte-Carlo Diffusion (MCD) and compare to differentiable AIS with ULA (Wu et al., 2020; Thin et al., 2021) and Unadjusted Hamiltonian Annealing (UHA) (Geffner & Domke, 2021; Zhang et al., 2021). We first investigate the performance of these approaches when the initial distribution is fixed. Next we explore the performance of the methods where the step-sizes, initial distribution, annealing

schedule, and per-timestep transition densities are learned. Our method's run-time is approximately two times the ULA baseline. For this reason we provide comparisons between the methods with increasing number of sampler steps. All target distributions are normalized, i.e. $\log Z = 0$.

Full experimental details, hyper-parameters, and model architectures can be found in Appendix B.

**Impact of reverse transition density.** We first study the impact of our score-based backward kernels compared to the standard AIS backward kernels. We sample from $\mathcal{N}(0, I)$ using initial distribution $\mathcal{N}(3, I)$. We use the same sampler, i.e. annealed ULA, for both methods using an increasing number of steps $K$. For both methods we learn the step-size per-timestep, trained to maximize the ELBO. Final $\log Z$ estimates are computed over 8192 samples. Results can be seen in Table 1 where we can clearly see the large impact our optimized backward kernels.

| Sampler | ULA | | | MCD (ours) | | |
|---------|-----|-----|-----|-----|-----|-----|
| # steps | 64 | 128 | 256 | 64 | 128 | 256 |
| Dim-20 | -0.83 $\pm$ 0.14 | -0.086 $\pm$ 0.121 | -0.059 $\pm$ 0.032 | -0.0013 $\pm$ 0.0046 | 0.0019 $\pm$ 0.0033 | -0.0002 $\pm$ 0.0008 |
| Dim-200 | -42.84 $\pm$ 1.49 | -17.10 $\pm$ 0.79 | -5.95 $\pm$ 0.59 | -0.087 $\pm$ 0.043 | -0.018 $\pm$ 0.026 | -0.020 $\pm$ 0.010 |
| Dim-500 | -142.40 $\pm$ 2.04 | -59.30 $\pm$ 4.30 | -24.57 $\pm$ 1.37 | -0.58 $\pm$ 0.15 | -0.045 $\pm$ 0.155 | -0.040 $\pm$ 0.065 |

Table 1: $\log Z$ estimates for annealing between $\mathcal{N}(0, 1)$ and $\mathcal{N}(3, 1)$. Averages and standard errors over 5 seeds.

**Full Differentiability.** We next estimate $Z$ for a more challenging distribution; a Gaussian mixture with 8 modes. Each mode's mean is drawn from $\mathcal{N}(0, 3I)$ and has covariance $I$. As in prior work (Geffner & Domke, 2021; Zhang et al., 2021), we take advantage of the fact that our importance weights are completely differentiable and we learn the initial distribution's mean and variance, the annealing schedule, and per-timestep transition densities. Results can be found in Table 2. In all settings our approach outperforms the baselines. In the 500-dimensional example, we find that the initial distribution learned by ULA and UHA collapses around a single mode. We do not observe this behavior with our method. We attribute this behavior to the reduced variance of the objective for our method.

| Sampler | ULA | | UHA | | MCD (ours) | |
|---------|-----|-----|-----|-----|-----|-----|
| # steps | 128 | 256 | 128 | 256 | 128 | 256 |
| Dim-20 | -0.68 $\pm$ 0.18 | -0.018 $\pm$ 0.584 | -0.50 $\pm$ 0.07 | -0.31 $\pm$ 0.05 | 0.0009 $\pm$ 0.0153 | 0.015 $\pm$ 0.013 |
| Dim-200 | -1.14 $\pm$ 0.01 | -1.70 $\pm$ 0.55 | -1.07 $\pm$ 0.37 | -0.41 $\pm$ 0.10 | -0.13 $\pm$ 0.06 | 0.040 $\pm$ 0.050 |
| Dim-500 | -2.97 $\pm$ 0.01 | -2.97 $\pm$ 0.01 | -2.98 $\pm$ 0.00 | -2.97 $\pm$ 0.00 | -1.50 $\pm$ 0.37 | -0.29 $\pm$ 0.10 |

Table 2: $\log Z$ estimates for mixtures of Gaussians. Averages and standard errors over 5 seeds.

**Normalizing Flow Evaluation.** Finally, we train NICE (Dinh et al., 2014) flows on the MNIST dataset. We train on 3 variants: the original $28 \times 28$ images, as well as images down-sampled to $14$ and $7 \times 7$. All models are trained for 100K steps and then $\log Z$ is estimated using 4096 importance samples. Results can be seen in Table 3. In the largest setting we can see that UHA outperforms ULA, but our method outperforms both.

| Dimension | ULA | UHA | MCD (ours) |
|-----------|-----|-----|------------|
| $7 \times 7$ | -0.14 | -0.17 | **-0.11** |
| $14 \times 14$ | -13.24 | -15.04 | **-6.25** |
| $28 \times 28$ | -141.29 | -82.16 | **-23.10** |

Table 3: $\log Z$ estimates for Normalizing flows.

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

# A PROOFS OF PROPOSITIONS

## A.1 PROOF OF PROPOSITION 1

*Proof.* The chain rule for the Kullback–Leibler divergence $D_{\mathrm{KL}}(Q||P)$ yields

$$D_{\mathrm{KL}}(Q||P) = D_{\mathrm{KL}}(q_K||\pi_K) + \mathbb{E}_{q_K}\Big[D_{\mathrm{KL}}(Q(\cdot|x_K)||P(\cdot|x_K))\Big], \tag{7}$$

where, from (1) and (2), the conditional distributions of $x_{1:k}$ given $x_0$ are equal to

$$Q(x_{0:K-1}|x_K) = \prod_{k=0}^{K-1} B_k^{\mathrm{opt}}(x_k|x_{k+1}), \qquad P(x_{0:K-1}|x_K) = \prod_{k=0}^{K-1} B_k(x_k|x_{k+1}), \tag{8}$$

The expression for $Q$ above follows directly from its time-reversed decomposition; i.e.

$$Q(x_{0:K}) = q_K(x_k) \prod_{k=0}^{K-1} \frac{q_k(x_k)F_{k+1}(x_{k+1}|x_k)}{q_{k+1}(x_{k+1})} = q_K(x_k) \prod_{k=0}^{K-1} B_k^{\mathrm{opt}}(x_k|x_{k+1}). \tag{9}$$

It thus follows directly from (7) and (8) that the backward transition kernels $(B_k)_{k=0}^{K-1}$ minimizing $D_{\mathrm{KL}}(Q||P)$ are $(B_k^{\mathrm{opt}})_{k=0}^{K-1}$ as this implies $P(x_{0:K-1}|x_K) = Q(x_{0:K-1}|x_K)$.

The variance decomposition formula yields for all $P$

$$\begin{aligned}
\mathrm{var}_Q[w(x_{0:K})] &= \mathrm{var}_{q_K}[\mathbb{E}_{Q(\cdot|x_K)}[w(x_{0:K})]] + \mathbb{E}_{q_K}[\mathrm{var}_{Q(\cdot|x_K)}[w(x_{0:K})]] \\
&= \mathrm{var}_{q_K}[w_{\mathrm{mar}}(x_K)] + \mathbb{E}_{q_K}[\mathrm{var}_{Q(\cdot|x_K)}[w(x_{0:K})]] \\
&\geq \mathrm{var}_{q_K}[w_{\mathrm{mar}}(x_K)].
\end{aligned}$$

By direct calculations, we also have $w_{\mathrm{mar}}(x_K) = \Gamma^{\mathrm{opt}}(x_{0:K})/Q(x_{0:K})$ so $P_{\mathrm{opt}}$ minimizes the variance of the evidence estimate. $\qquad\square$

## A.2 PROOF OF PROPOSITION 2

We establish first here Proposition 3 and Proposition 4. Both results can then be easily combined to obtain Proposition 2.

**Proposition 3.** *Under regularity conditions, one has*

$$D_{\mathrm{KL}}(\mathcal{Q}||\mathcal{P}_\theta) = \mathbb{E}_\mathcal{Q}\Big[\int_0^T ||s_\theta(t, X_t) - \nabla \log q_t(X_t)||^2 \mathrm{d}t\Big] + C_1 \tag{10}$$

$$= \sum_{k=1}^K \int_{t_{k-1}}^{t_k} \mathbb{E}_\mathcal{Q}\big[||s_\theta(t, X_t) - \nabla \log q_{t|t_{k-1}}(X_t|X_{t_{k-1}})||^2\big]\, \mathrm{d}t + C_2, \tag{11}$$

*for constants $C_1, C_2$ independent of $\theta$, where $t_k = k\delta$ with $K = T/\delta$ and $q_{t|s}(x'|x)$ is the density of $X_t = x'$ given $X_s = x$ under $\mathcal{Q}$.*

To establish (10), we follow arguments similar to (Song et al., 2021a, Theorem 2). The loss (11) we then consider differs from the one uses in the score-based generative modeling literature. This is because, contrary to the Ornstein–Ulhenbeck process used for generative modeling, the transition

density $q_{t'|t}(x'|x)$ of the forward diffusion (4) is not available in closed-form and can only be approximated reliably when $t' - t$ is small. Practically, to obtain a tractable criterion, we need to first approximate the integrals in (11) by the rectangular rule. We also discretize the Langevin dynamics using an Euler–Maruyama scheme; i.e. we use an approximation $Q$ of $\mathcal{Q}$ based on the ULA kernel $F_k(x'|x) = \mathcal{N}(x'; x + \delta\nabla\log\pi_{t_k}(x); 2\delta I)$ approximating $q_{t_k|t_{k-1}}(x'|x)$. We thus finally obtain a loss

$$\mathcal{L}(\theta) = \delta\sum_{k=1}^{K}\mathbb{E}_Q\left[||s_\theta(k, x_k) - \nabla\log F_k(x_k|x_{k-1})||^2\right]. \tag{12}$$

*Proof.* We assume here that $\pi_t(x)$ and $s_\theta(t, x)$ are sufficiently regular to yield a unique (weak) solution of the SDEs. By the chain rule for KL divergences, one has

$$D_{\mathrm{KL}}(\mathcal{Q}||\mathcal{P}_\theta) = D_{\mathrm{KL}}(q_T||\pi_T) + \mathbb{E}_{q_T}\left[D_{\mathrm{KL}}(\mathcal{Q}(\cdot|X_T)||\mathcal{P}_\theta(\cdot|X_T))\right] \tag{13}$$

where $\mathcal{Q}(\cdot|X_T))$ and $\mathcal{P}_\theta(\cdot|X_T)$ are the path measures induced by

$$\mathrm{d}Y_t = \left\{-\nabla\log\pi_{T-t}(Y_t) + 2\nabla\log q_{T-t}(Y_t)\right\}\mathrm{d}t + \sqrt{2}\mathrm{d}B_t, \qquad Y_0 = X_T, \tag{14}$$

and

$$\mathrm{d}Y_t = \left\{-\nabla\log\pi_{T-t}(Y_t) + 2\nabla\log s_\theta(T-t, Y_t)\right\}\mathrm{d}t + \sqrt{2}\mathrm{d}B_t, \qquad Y_0 = X_T. \tag{15}$$

We now use Girsanov theorem (see e.g. (Klebaner, 2012, Section 10.3)) to compute the Radon–Nikodym derivative $\mathrm{d}Q(\cdot|X_T)/\mathrm{d}\mathcal{P}_\theta(\cdot|X_T)$ so that

$$\mathbb{E}_{q_T}\left[D_{\mathrm{KL}}(\mathcal{Q}(\cdot|X_T)||\mathcal{P}_\theta(\cdot|X_T))\right]$$

$$= -\mathbb{E}_Q\left[\log\frac{\mathrm{d}\mathcal{P}_\theta(\cdot|X_T)}{\mathrm{d}Q(\cdot|X_T)}\right]$$

$$= \mathbb{E}_Q\left[\sqrt{2}\int_0^T(\nabla\log q_t(X_t) - s_\theta(t, X_t))\mathrm{d}B_t + \int_0^T||\nabla\log q_t(X_t) - s_\theta(t, X_t)||^2\mathrm{d}t\right]$$

$$= \mathbb{E}_Q\left[\int_0^T||\nabla\log q_t(X_t) - s_\theta(t, X_t)||^2\mathrm{d}t\right],$$

as $\mathbb{E}_Q\left[\int_0^T f_t(X_t)\mathrm{d}B_t\right] = 0$ for any function $f_t$.

As in (De Bortoli et al., 2021) in a different context, we can write for any partition of $[0, T]$ defined by $t_0 = 0 < t_1 < \cdots < t_{K-1} < t_K = T$

$$\mathbb{E}_Q\left[\int_0^T||\nabla\log q_t(X_t) - s_\theta(t, X_t)||^2\mathrm{d}t\right] = \int_0^T\int||\nabla\log q_t(x) - s_\theta(t, x)||^2 q_t(x)\mathrm{d}x\mathrm{d}t$$

$$= \sum_{k=1}^{K}\int_{t_{k-1}}^{t_k}\int||\nabla\log q_t(x) - s_\theta(t, x)||^2 q_t(x)\mathrm{d}x\mathrm{d}t$$

where, for a constant $c$ independent of $\theta$, we have

$$\int_{t_{k-1}}^{t_k}\int||\nabla\log q_t(x) - s_\theta(t, x)||^2 q_t(x)\mathrm{d}x\mathrm{d}t$$

$$= \int_{t_{k-1}}^{t_k}\int\left\{||\nabla\log q_t(x)||^2 + ||s_\theta(t, x)||^2 - 2s_\theta(t, x)^T\nabla\log q_t(x)\right\}q_t(x)\mathrm{d}x\mathrm{d}t$$

$$= \int_{t_{k-1}}^{t_k}\int\left\{||s_\theta(t, x)||^2 - 2s_\theta(t, x)^T\nabla\log q_t(x)\right\}q_t(x)\mathrm{d}x\mathrm{d}t + c.$$

Now we have

$$\int_{t_{k-1}}^{t_k}\int s_\theta(t, x)^\mathrm{T}\nabla\log q_t(x)q_t(x)\mathrm{d}x\mathrm{d}t = \int_{t_{k-1}}^{t_k}\int s_\theta(t, x)^\mathrm{T}\nabla q_t(x)\mathrm{d}x\mathrm{d}t \tag{16}$$

where, using Chapman-Kolmogorov, $q_t$ satisfies

$$q_t(x) = \int q_{t_{k-1}}(x_{t_{k-1}}) q_{t|t_{k-1}}(x|x_{t_{k-1}}) \mathrm{d}x_{t_{k-1}}. \tag{17}$$

It follows that

$$\nabla q_t(x) = \int q_{t_{k-1}}(x_{t_{k-1}}) \nabla q_{t|t_{k-1}}(x|x_{t_{k-1}}) \mathrm{d}x_{t_{k-1}}. \tag{18}$$

Hence, we have

$$\int_{t_{k-1}}^{t_k} \int s_\theta(t,x)^{\mathrm{T}} \nabla q_t(x) \mathrm{d}x \mathrm{d}t$$

$$= \int_{t_{k-1}}^{t_k} \int \int s_\theta(t,x)^{\mathrm{T}} \nabla \log q_{t|t_{k-1}}(x|x_{t_{k-1}}) q_{t_{k-1}}(x_{t_{k-1}}) q_{t|t_{k-1}}(x|x_{t_{k-1}}) \mathrm{d}x_{t_{k-1}} \mathrm{d}x \mathrm{d}t$$

so minimizing $\mathbb{E}_{q_T}\left[ D_{\mathrm{KL}}(\mathcal{Q}(\cdot|X_T)||\mathcal{P}_\theta(\cdot|X_T)) \right]$ w.r.t. $\theta$ is equivalent to minimize

$$\sum_{k=1}^{K} \int_{t_{k-1}}^{t_k} \int \int ||s_\theta(t,x)||^2 q_{t_{k-1}}(x_{t_{k-1}}) q_{t|t_{k-1}}(x|x_{t_{k-1}}) \mathrm{d}x_{t_{k-1}} \mathrm{d}x \mathrm{d}t$$

$$-2 \sum_{k=1}^{K} \int_{t_{k-1}}^{t_k} \int \int s_\theta(t,x)^{\mathrm{T}} \nabla \log q_{t|t_{k-1}}(x|x_{t_{k-1}}) q_{t_{k-1}}(x_{t_{k-1}}) q_{t|t_{k-1}}(x|x_{t_{k-1}}) \mathrm{d}x_{t_{k-1}} \mathrm{d}x \mathrm{d}t$$

$$= \sum_{k=1}^{K} \int_{t_{k-1}}^{t_k} \int \int ||s_\theta(t,x) - \nabla \log q_{t|t_{k-1}}(x|x_{t_{k-1}})||^2 q_{t_{k-1}}(x_{t_{k-1}}) q_{t|t_{k-1}}(x|x_{t_{k-1}}) \mathrm{d}x_{t_{k-1}} \mathrm{d}x \mathrm{d}t + C$$

where $C$ is independent of $\theta$. Hence, this is equivalent to minimizing (11). $\qquad\square$

We now establish results about the discrete-time Kullback–Leibler divergence $D_{\mathrm{KL}}(Q||P_\theta)$. First note that

$$D_{\mathrm{KL}}(Q||P_\theta) = \mathbb{E}_Q\left[ \log \frac{Q(x_{0:K})}{P_\theta(x_{0:K})} \right]$$

$$= \mathbb{E}_Q\left[ \log \frac{\pi_0(x_0) \prod_{k=0}^{K-1} F_{k+1}(x_{k+1}|x_k)}{\pi_K(x_K) \prod_{k=0}^{K-1} B_k^\theta(x_k|x_{k+1})} \right]$$

$$= -\sum_{k=0}^{K-1} \mathbb{E}_Q\left[ \log B_k^\theta(x_k|x_{k+1}) \right] + C_1, \tag{19}$$

where, as $B_k^\theta(x'|x) = \mathcal{N}(x'; x - \delta \nabla \log \pi_{k+1}(x) + 2\delta s^\theta(k+1,x), 2\delta I)$, one has

$$-\log B_k^\theta(x_k|x_{k+1}) = \frac{1}{4\delta} ||x_k - x_{k+1} + \delta \nabla \log \pi_{k+1}(x_{k+1}) - 2\delta s^\theta(k+1, x_{k+1})||^2 + C_2$$

$$= \delta \left\| s^\theta(k+1, x_{k+1}) - \frac{1}{2\delta}(x_k - x_{k+1} + \delta \nabla \log \pi_{k+1}(x_{k+1})) \right\|^2 + C_2 \quad (20)$$

$$\approx \delta \left\| s^\theta(k+1, x_{k+1}) - \frac{1}{2\delta}(x_k - x_{k+1} + \delta \nabla \log \pi_{k+1}(x_k)) \right\|^2 + C_2$$

$$= \delta \left\| s^\theta(k+1, x_{k+1}) - \nabla \log F_{k+1}(x_{k+1}|x_k) \right\|^2 + C_2, \tag{21}$$

where we have used $\pi_{k+1}(x_{k+1}) \approx \pi_{k+1}(x_k)$ for $\delta \ll 1$. The sum over $k = 0, ..., K-1$ of the first terms on the r.h.s. of (20) are equal to the loss $\mathcal{L}(\theta)$ defined in (12). More rigorously, we can prove the following result.

**Proposition 4.** *Under Lipschitz assumptions on $(\nabla \log \pi_k)_{k=1}^K$ and moment assumptions on the scores approximations and their derivative w.r.t. $\theta$, the gradient of the Kullback–Leibler divergence $D_{\mathrm{KL}}(Q||P_\theta)$ satisfies*

$$\nabla D_{\mathrm{KL}}(Q||P_\theta) = \nabla \mathcal{L}(\theta) + \epsilon(\theta), \tag{22}$$

*for $\mathcal{L}(\theta)$ defined in (12) and a function $\epsilon$ satisfying $\lim_{K \to \infty} \epsilon(\theta) = 0$.*

*Proof.* In the rest of the proof, all the expectations are taken w.r.t. $Q$ unless mentioned otherwise and we drop it from the notations for simplicity. However as we take gradients w.r.t. to both $x$ and $\theta$, this is indicated notationally to avoid confusion. We also assume that $\theta$ is a scalar in the proof, the extension to the multivariate case is straightforward.

Using (19), we have

$$\nabla_\theta D_{\mathrm{KL}}(Q||P_\theta) = -\sum_{k=0}^{K-1} \mathbb{E}\big[\nabla_\theta \log B_k^\theta(x_k|x_{k+1})\big], \tag{23}$$

where, from (20), one has

$$-\nabla_\theta \log B_k^\theta(x_k|x_{k+1}) \tag{24}$$

$$=\delta\nabla_\theta\Big\|s^\theta(k+1, x_{k+1}) - \frac{1}{2\delta}(x_k - x_{k+1} + \delta\nabla_x \log \pi_{k+1}(x_{k+1}))\Big\|^2$$

$$=2\delta\nabla_\theta s^\theta(k+1, x_{k+1})^{\mathrm{T}}(s^\theta(k+1, x_{k+1}) - \frac{1}{2\delta}(x_k - x_{k+1} + \delta\nabla_x \log \pi_{k+1}(x_{k+1}))). \tag{25}$$

We also have

$$\nabla_\theta \mathcal{L}(\theta) = \delta \sum_{k=0}^{K-1} \mathbb{E}\left[\nabla_\theta\Big\|s_\theta(k, x_k) - \nabla_x \log F_k(x_k|x_{k-1})\Big\|^2\right], \tag{26}$$

where

$$\delta\nabla_\theta\Big\|s_\theta(k, x_k) - \nabla_x \log F_k(x_k|x_{k-1})\Big\|^2$$

$$=\delta\nabla_\theta\Big\|s_\theta(k, x_k) - \frac{1}{2\delta}(x_k - x_{k+1} + \delta\nabla_x \log \pi_{k+1}(x_k))\Big\|^2$$

$$=2\delta\nabla_\theta s^\theta(k+1, x_{k+1})^{\mathrm{T}}(s^\theta(k+1, x_{k+1}) - \frac{1}{2\delta}(x_k - x_{k+1} + \delta\nabla_x \log \pi_{k+1}(x_k))). \tag{27}$$

So we obtain by using (19) and (20)

$$\nabla_\theta D_{\mathrm{KL}}(Q||P_\theta) = \nabla_\theta \mathcal{L}(\theta) + \epsilon(\theta), \tag{28}$$

for

$$\epsilon(\theta) = 2\delta\mathbb{E}\left[\sum_{k=0}^{K-1} \nabla_\theta s^\theta(k+1, x_{k+1})^{\mathrm{T}}(\nabla_x \log \pi_{k+1}(x_k) - \nabla_x \log \pi_{k+1}(x_{k+1}))\right]. \tag{29}$$

Hence we have

$$|\epsilon(\theta)| \leq 2\delta \sum_{k=0}^{K-1} \mathbb{E}\left[|\nabla_\theta s^\theta(k+1, x_{k+1})^{\mathrm{T}}(\nabla_x \log \pi_{k+1}(x_k) - \nabla_x \log \pi_{k+1}(x_{k+1}))|\right]$$

$$\leq 2\delta \sum_{k=0}^{K-1} \mathbb{E}\left[\Big\|\nabla_\theta s^\theta(k+1, x_{k+1})\Big\|^2\right]^{1/2} \mathbb{E}\left[\Big\|\nabla_x \log \pi_{k+1}(x_k) - \nabla_x \log \pi_{k+1}(x_{k+1})\Big\|^2\right]^{1/2} \tag{30}$$

As we assume that the gradients $\nabla_x \log \pi_{k+1}$ are $L$-Lipschitz, then

$$\mathbb{E}\left[\Big\|\nabla_x \log \pi_{k+1}(x_k) - \nabla_x \log \pi_{k+1}(x_{k+1})\Big\|^2\right] \leq L^2\mathbb{E}\left[\Big\|x_{k+1} - x_k\Big\|^2\right]$$

$$\leq 2L^2\delta\mathbb{E}\left[\delta\Big\|\nabla_x \log \pi_{k+1}(x_k)\Big\|^2 + 2M\right], \tag{31}$$

where $M = \mathbb{E}_{Z\sim\mathcal{N}(0,I)}[||Z||^2]$ as $x_{k+1} = x_k + \delta\nabla_x \log \pi_{k+1}(x_k) + \sqrt{2\delta}Z$ under $Q$. Hence if we assume that the following moment assumptions hold

$$\limsup_K \max_{k=0,...,K-1} \mathbb{E}_{Q_K}\left[\Big\|s^\theta(k+1, x_{k+1})\Big\|^2\right] \leq E \tag{32}$$

and

$$\limsup_K \max_{k=0,...,K-1} \mathbb{E}_{Q_K} \left[ \left\| \nabla_\theta s^\theta(k+1, x_{k+1}) \right\|^2 \right] \leq E, \tag{33}$$

for some constant $E < \infty$, where we have emphasized here notationally that $Q$ is a function of $K$, then we obtain from (30), (31), (32), (33) that $\epsilon(\theta) = O(\sqrt{\delta})$ and the result follows.

$\square$

## B  EXPERIMENTAL DETAILS

### B.1  SAMPLER PARAMETERIZATION

For all models, the step size was learned via a function $\epsilon_\theta(t)$ which is a 2-layer neural network with 32 hidden units, followed by a scaled sigmoid function which constrains $\epsilon_\theta(t) < .25$. As in prior work (Geffner & Domke, 2021) we found this alleviated some instabilities in training.

When learning the annealing schedule, we parameterize an increasing sequence of $T$ steps using unconstrained parameters $b_t$ (initialized to the same constant). We map these to our annealing schedule with

$$\beta_t = \frac{\sum_{t' \leq t} \sigma(b_{t'})}{\sum_{t'=1}^T \sigma(b_{t'})} \tag{34}$$

where we fix $\beta_0 = 0$ and $\sigma$ is the sigmoid function. This ensures that $\beta_0 = 0$, $\beta_K = 1$, and $\beta_t < \beta_{t'}$ when $t < t'$.

For UHA (Geffner & Domke, 2021), we also learn the momentum refreshment parameter $\eta \in (0, 1)$. We parameterize this with a parameter $u$ and define $\eta = .98\sigma(u) + .01$ to keep the values in the range $(.01, .99)$ which we found alleviated training instabilities.

### B.2  SCORE MODEL PARAMETERIZATION

We parameterize our score model $s_\theta(t, x)$ using an MLP residual network. We first project the $x$ to dim $d_h$ using a linear layer and embed discrete time steps $t$ to dim $d_t$ using a learned embedding map. We then apply $k$ residual blocks.

Each block begins with a layer norm (Ba et al., 2016) operation followed by a nonlinearity. We project the hidden representation to dim $2 \cdot d_h$ using a linear layer, project the embedding of $t$ to dim $2 \cdot d_h$ using another linear map and add them together. We then apply another nonlinearity and then project the back to $d_h$ using another linear layer. We use the swish nonlinearity (Ramachandran et al., 2017) throughout.

To ensure our ELBO is initialized to a reasonable value we *warm start* it so that at initialization, the score model outputs the standard AIS backward kernels. We do this by defining a score model $\tilde{s}_\theta(x, t)$ as explained above (but set the final layer weights to 0 at initialization) and define:

$$s_\theta(x, t) = \tilde{s}_\theta(x, t) + \nabla_x \gamma_t(x) \tag{35}$$

which we found this led to much faster convergence and better results overall.

### B.3  HYPER-PARAMETERS

In all experiments we use $k = 3$ residual blocks. For our Gaussian experiments we set $d_h = 128$ and $d_t = 16$. For our Gaussian Mixture experiments we set $d_h = 512$ and $d_t = 32$. For our flow experiments we set $d_h$ to 128, 256, and 512 for image sizes $7 \times 7$, $14 \times 14$, and $28 \times 28$, respectively. For all flows we set $d_t = 32$.

All models are trained with the Adam optimizer Kingma & Ba (2014) with learning rate 0.001. We train for 100,000 iterations and estimate $\log Z$ using 8912 importance samples. For the Gaussian and the Gaussian Mixture experiments, we present average performance (with standard error) over 5 different random seeds.

