# OpenReview forum: "Annealed Importance Sampling meets Score Matching"
_ICLR.cc/2022/Workshop/DGM4HSD — ICLR 2022 DGM4HSD workshop Poster_

### Official Review · Reviewer_npnL · 2022-03-23
**Improving AIS with a better extended target distribution**

**Rating:** 7
**Confidence:** 3

**Review:**

This paper proposes a novel AIS procedure for Bayesian evidence estimation. The central idea is that, while the extended target distribution used by standard AIS (eq 2) is convenient, it is not optimal. The authors suggest a better extended target distribution motivated by score matching. On the three benchmark examples, the proposed MCD method outperforms the existing AIS methods.

I think the presentation could be more detailed but I understand the authors need to observe the page limit. Below are some questions on which I suggest the authors add more explanation:

1. Is $P^{\rm opt}$ the same as $Q$, assuming that the Markov transitions are well-mixed?

2. Does minimizing $D_{KL}(Q||P_{\theta})$ means maximizing $\Gamma_{\theta}$ on samples generated from the forward proposal $Q$? It would be good to have an algorithm box to clarify the actual implementation of the proposed method.

3. If one continues to increase the number of integration steps for UHA, would it be comparable with MCD in terms of accuracy? If so, which method is more efficient to reach the same accuracy?

4. In Table 3 the results are reported in terms of absolute errors; why not just report the actual $\log Z$, like the previous examples? It's known that AIS tends to underestimate the evidence. Here is MCD overestimating or underestimating?

---

### Decision · Program_Chairs · 2022-03-26

Accept (Poster)